# Pleiotropic requirements for human TDP-43 in the regulation of cell and organelle homeostasis

Agnes Roczniak-Ferguson[1,2,3], Shawn M Ferguson[1,2,3]

**TDP-43 is an RNA-binding protein that forms cytoplasmic aggregates in multiple neurodegenerative diseases. Although the loss of normal TDP-43 functions likely contributes to disease pathogenesis, the cell biological consequences of human TDP-43 depletion are not well understood. We, therefore, generated human TDP-43 knockout (KO) cells and subjected them to parallel cell biological and transcriptomic analyses. These efforts yielded three important discoveries. First, complete loss of TDP-43 resulted in widespread morphological defects related to multiple organelles, including Golgi, endosomes, lysosomes, mitochondria, and the nuclear envelope. Second, we identified a new role for TDP-43 in controlling mRNA splicing of Nup188 (nuclear pore protein). Third, analysis of multiple amyotrophic lateral sclerosis causing TDP-43 mutations revealed a broad ability to support splicing of TDP-43 target genes. However, as some TDP-43 disease-causing mutants failed to fully support the regulation of specific target transcripts, our results raise the possibility of mutation-specific loss-of-function contributions to disease pathology.**

## Introduction

Transactivation response element DNA-binding protein 43 (*TARDBP*, also known as *TDP-43*) belongs to the family of heterogeneous nuclear ribonucleoproteins (Purice & Taylor, 2018). Mutations in the *TARDP* gene cause a familial form of amyotrophic lateral sclerosis (ALS) (Kabashi et al, 2008; Sreedharan et al, 2008) that is accompanied by the formation of neuronal cytoplasmic TDP-43 inclusions (Neumann et al, 2006). TDP-43 inclusions also occur in familial forms of ALS and frontotemporal dementia (FTD) that are caused by mutations in other genes as well as in sporadic forms of these and other neurodegenerative diseases (Amador-Ortiz et al, 2007; Rademakers et al, 2012; Ling et al, 2013; Mackenzie & Neumann, 2016; Ayaki et al, 2018). Cytoplasmic TDP-43 aggregates also occur in muscle in the context of inclusion body myopathy (Weihl et al, 2008).

The strong genetic and pathological links between TDP-43 and neurodegenerative disease have stimulated intense interest in elucidating the relationships between its normal and pathological functions (Taylor et al, 2016). Although TDP-43 was originally identified and named for its ability to bind to HIV-1 long terminal repeat DNA, it is now understood that TDP-43 is ubiquitously expressed in all cell types and plays an important physiological role in regulating the splicing of multiple endogenous human mRNAs (Tollervey et al, 2011; Ling et al, 2015; Appocher et al, 2017; Conlon & Manley, 2017). The specific RNA targets for TDP-43 vary between species. However, there is a conserved role for TDP-43 in suppressing the inclusion of cryptic exons via binding to UG dinucleotide repeats in their flanking regions (Chiang et al, 2010; Polymenidou et al, 2011; Sephton et al, 2011; Lukavsky et al, 2013; Ling et al, 2015; Tan et al, 2016). The loss of such activity results in the production of numerous frameshifted transcripts that are frequently targets of nonsense-mediated decay. Identifying human genes affected by cryptic exon insertion arising from TDP-43 dysfunction and understanding the consequences of their disruption is thus important for understanding both the normal mechanisms whereby TDP-43 ensures splicing fidelity as well as the contributions of aberrant mRNA splicing to disease pathology. In addition to regulating mRNA splicing, TDP-43 has also been implicated in the regulation of other aspects of RNA biology including, transcription, microRNA processing, RNA stability, and regulation of cytoplasmic RNP complexes such as stress granules, myogranules involved in muscle regeneration, and granules involved in axonal RNA transport in neurons (Ratti & Buratti, 2016; Gopal et al, 2017; Vogler et al, 2018).

Efforts to define TDP-43 function in mice through knockout (KO) strategies revealed that TDP-43 is absolutely required for embryonic development and viability (Chiang et al, 2010; Kraemer et al, 2010; Sephton et al, 2010; Wu et al, 2010). Even TDP-43 conditional KO strategies in specific cell types resulted in proliferation defects and/or cell death (Chiang et al, 2010). The lethality arising from TDP-43 depletion has limited efforts to define both normal TDP-43 functions as well as the cell biological consequences of TDP-43 depletion. As a result of these challenges, the disease contributions of nuclear TDP-43 depletion and/or TDP-43 inactivation associated

[1]Department of Cell Biology, Yale University School of Medicine, New Haven, CT, USA  [2]Department of Neuroscience, Yale University School of Medicine, New Haven, CT, USA  [3]Program in Cellular Neuroscience, Neurodegeneration and Repair, Yale University School of Medicine, New Haven, CT, USA

Correspondence: shawn.ferguson@yale.edu

with its cytoplasmic aggregation remain uncertain. Results from mouse studies are further complicated by the lack of conservation in TDP-43 targets between species (Prudencio et al, 2012; Ling et al, 2015). Studies in human cells where TDP-43 has been partially depleted (but not eliminated) by RNAi approaches have identified specific targets related to the functions of several organelles/pathways including autophagy and nuclear import (Ling et al, 2015; Stalekar et al, 2015; Prpar Mihevc et al, 2016; Xia et al, 2016). Although these results are intriguing, it remains unclear to what extent the regulation of any single TDP-43 target contributes to the total influence of TDP-43 on cell physiology.

As a comprehensive understanding of TDP-43 functions is critical for understanding normal human cell biology as well as for deciphering disease mechanisms, we have developed the first human TDP-43 KO cells and used them to perform comprehensive cell biological and transcriptomic analyses of the consequences of TDP-43 depletion. The results of these experiments revealed that TDP-43 is required for the homeostasis of multiple subcellular organelles. Transcriptomic analysis of TDP-43 KO cells both confirmed the impact of TDP-43 on multiple known targets but also revealed new candidates. Given recent interest in the contributions of nuclear transport defects to neurodegenerative diseases associated with TDP-43 pathology (Ward et al, 2014; Gao et al, 2017; Kim & Taylor, 2017; Chou et al, 2018; Zhang et al, 2018), we highlight in particular the identification of the nucleoporin, Nup188, as a novel target of TDP-43–dependent splicing regulation. Furthermore, our analysis of the ability of multiple disease-causing TDP-43 mutants to rescue TDP-43 KO phenotypes supports the general functionality of these proteins. These results are consistent with a model wherein TDP-43 loss of function arising from cytoplasmic accumulation and aggregation of mutant TDP-43 proteins rather than a direct loss of their fundamental ability to support RNA causes them to confer disease risk. However, the fact that we observed some TDP-43 mutants were unable to efficiently support the regulation of specific transcripts raises the possibility of unidentified, disease-relevant, transcripts that are especially sensitive to TDP-43 mutations.

Collectively, this study takes advantage of a newly generated TDP-43 KO human cell line to shed new light on cellular functions of TDP-43 and establishes such cells as a platform for investigating the functionality of disease-causing TDP-43 mutations.

## Results and Discussion

### Generation of a TDP-43 KO cell culture model

Efforts to define TDP-43 function through genetic depletion approaches have previously been implemented in mice as well as in human cells in culture (Chiang et al, 2010; Kraemer et al, 2010; Sephton et al, 2010; Wu et al, 2010; Ling et al, 2015; Stalekar et al, 2015; Prpar Mihevc et al, 2016; Xia et al, 2016). Although these strategies have been informative in identifying many mRNA transcripts whose splicing depends on TDP-43 and physiological processes that are impaired in response to reduced TDP-43 levels, the essential functions of human TDP-43 have not been defined via a complete

KO. Furthermore, as the splicing targets of TDP-43 are highly species dependent, insights obtained from mouse studies have not always yielded results that were translatable to humans (Prudencio et al, 2012; Ling et al, 2015). These factors motivated us to generate human TDP-43 KO cells via CRISPR-Cas9–mediated genome editing.

As past efforts to deplete TDP-43 reached the conclusion that this gene is essential and that its complete absence is not compatible with life, we first focused on HeLa cells to test the feasibility of a TDP-43 KO approach in human cells as this cell line is both highly adapted to robust growth in culture and very amenable to CRISPR-Cas9–mediated genome editing. Following expression of Cas9 and two distinct TDP-43–targeted small guide RNAs (sgRNAs), either alone or in combination, we analyzed polyclonal cell populations by immunoblotting and observed significant depletion of TDP-43 protein levels (Fig 1A). Clonal populations of these cells were next isolated and screened by immunoblotting to identify clones that lacked the TDP-43 protein (Fig 1B). The sgRNAs used for generating these KO cells target exon 3 (out of eight exons in total), which encodes part of RNA Recognition Motif 1 (RRM1) near the beginning of the TDP-43 protein. Sequencing of this region of genomic DNA confirmed the loss of wildtype (WT) *TDP-43* sequence at the Cas9 cut site in all copies of the *TARDBP* gene and the presence of small insertions and deletions that result in frameshifts in our TDP-43 KO cell line (Fig S1A–C). In addition to prematurely terminating translation, the presence of frameshift mutations within this early exon is predicted to result in depletion of the TDP-43 transcript by nonsense-mediated decay. Consistent with the conclusion that we successfully eliminated production of the TDP-43 protein, immunoblotting with two distinct antibodies (including one that recognizes the N terminus) confirmed the loss of the 43-kD TDP-43 protein and the absence of any smaller fragments (Fig S1D and E). To generate a control for subsequent investigation of the phenotypes arising from loss of TDP-43, we next generated a "TDP-43 rescue" line by transducing the KO cells with a lentivirus encoding a wild-type human TDP-43 cDNA (Fig 1C). It is striking that this rescue strategy where TDP-43 expression was driven by the strong CMV promoter yielded TDP-43 protein levels that so closely matched the endogenous expression levels of TDP-43. It is not clear whether this is serendipitous or whether it reflects a homeostatic mechanism whereby cells tightly control TDP-43 protein levels. Although TDP-43 auto-regulatory mechanisms have previously been described (Ayala et al, 2011; Avendano-Vazquez et al, 2012), the transgene used here lacks the key intronic and UTR sequences that were previously shown to underlie known auto-regulatory mechanisms. Therefore, any homeostatic control of TDP-43 protein levels in these cells would have to arise via alternative mechanisms.

These DNA sequencing and immunoblotting results established that we had successfully generated human TDP-43 KO cells. To further characterize these cells, we examined POLDIP3, a well-defined target of TDP-43–dependent splicing regulation (Fiesel et al, 2012; Shiga et al, 2012). The POLDIP3 protein migrates as two bands on SDS–PAGE gels that reflect the presence or absence of exon 3, whose inclusion is dependent on TDP-43 (Fiesel et al, 2012; Shiga et al, 2012). In WT cells, the POLDIP3 protein migrated primarily at the expected size of 46 kD, with only a small amount of the smaller exon 3 lacking variant of this protein (Fig 1D). In TDP-43 KO cells, however, POLDIP3 migrated predominantly as the smaller

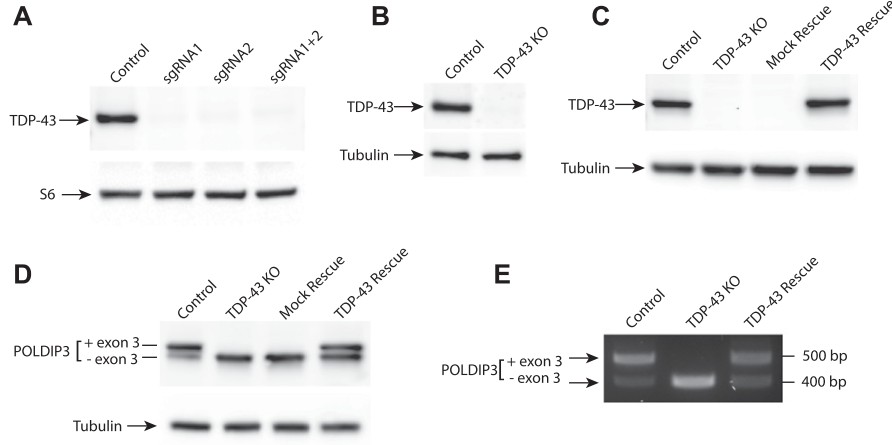

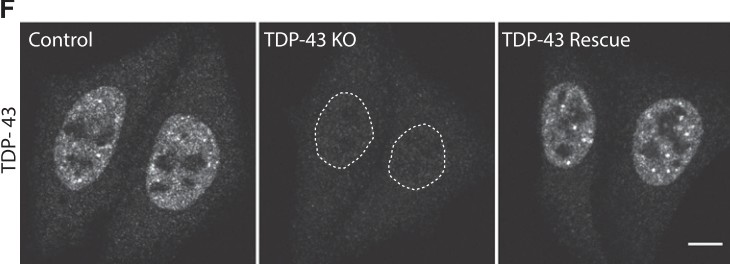

**Figure 1. Generation and validation of human TDP-43 KO cells.**
**(A, B)** Immunoblot analysis of polyclonal populations of HeLa cells transfected with Cas9 plus two sgRNAs against TDP-43 separately or in combination. Ribosomal protein S6 serves as a loading control (B) Immunoblot analysis of a clonal population of TDP-43 KO HeLa cells. **(C)** Immunoblot evaluation of the restoration of TDP-43 expression via lentiviral transduction of TDP-43 KO cells with an untagged, wild-type form of TDP-43. **(D)** Immunoblotting revealed that levels of the larger POLDIP3 isoform were rescued by re-expression of TDP-43 in the TDP-43 KO cells. **(E)** Assessment of POLDIP3 exon 3 inclusion by RT-PCR. **(F)** Confocal immunofluorescence microscopy shows similar nuclear localization of the TDP-43 protein in both parental HeLa cells and the rescue cell line. Scale bar = 10 μm.

form of the protein (Fig 1D). The specificity of this TDP-43 KO phenotype was demonstrated by observations that stably reintroducing TDP-43 back into the KO cells via lentiviral transduction reversed the POLDIP3 phenotype (Fig 1D). TDP-43–dependent regulation of *POLDIP3* mRNA splicing was further confirmed by RT-PCR analysis of RNA isolated from either WT, TDP-43 KO, or TDP-43–rescued KO cells (Fig 1E). Immunofluorescence analysis identified a nuclear TDP-43 signal that was selectively present only in the WT and rescued cell lines, whereas a minimal, nonspecific, nuclear, and cytoplasmic immunofluorescence signal was observed in the KO cells (Fig 1F).

## Broad cell biological consequences of the TDP-43 KO

Previous studies of TDP-43 function in human cells have broadly focused on widespread transcriptional changes or very specific phenotypes relating to a particular transcript and/or the organelle in which it functions (Ling et al, 2015; Stalekar et al, 2015; Prpar Mihevc et al, 2016; Xia et al, 2016). Although these approaches have yielded valuable insights, they left gaps with respect to understanding the full cellular impact of TDP-43 perturbations. We, thus, performed confocal microscopy analysis of cells that were stained in parallel with antibodies against established markers of multiple intracellular organelles. These efforts revealed a pleiotropic set of specific changes arising from the loss of TDP-43 that were reversed in the rescued cell line (Fig 2). Key changes that were observed include abnormal morphology of the nuclear envelope (Fig 2A), Golgi fragmentation (Fig 2B), early endosome dispersal (Fig 2C), and clustering of both lysosomes (Fig 2D) and mitochondria (Fig 2E) in

the perinuclear region. These morphological changes in organelles and their subcellular distribution were not accompanied by any overt changes in the steady-state organization of the microtubule and actin cytoskeletons (Fig S2A and B). These changes to the morphology and positioning of multiple organelles demonstrate the existence of pleiotropic cell biological changes that arise in response to loss of TDP-43 function.

In addition to their abnormal subcellular distribution, the lysosomes in TDP-43 KO cells lacked cathepsin L (a major lysosomal protease that supports protein degradation, Figs 3A and S3A). Cathepsin L localization to lysosomes was restored in TDP-43 rescue cells (Fig 3A). Cathepsin L normally undergoes proteolytic processing upon delivery to lysosomes that is essential for its activation. This maturation of cathepsin L was impaired in TDP-43 KO cells and was restored in the TDP-43 rescue cells (Fig 3D and E). Similarly to cathepsin L, cathepsin B processing in lysosomes of TDP-43 KO cells was also impaired (Fig 3D). Cathepsin D, on the other hand, still localized to the lysosomes in TDP-43 KO cells (Figs 3B and S3B), suggesting that distinct mechanisms exist for the lysosomal delivery of the various cathepsins. The maturation of the cathepsin D protein was likewise unaffected in TDP-43 KO cells (Fig 3D).

The lysosome defects in TDP-43 depleted cells were intriguing as mutations in the progranulin gene (GRN) give rise to FTD with TDP-43 pathology (Baker et al, 2006; Cruts et al, 2006) and the progranulin protein localizes to lysosomes (Hu et al, 2010; Gowrishankar et al, 2015; Holler et al, 2017). Interestingly, we observed that progranulin localization to lysosomes was greatly diminished in the absence of TDP-43 (Figs 3C and S3C). This was accompanied by a build-up of full-length progranulin inside the cells (Fig 3D and F), an effect that

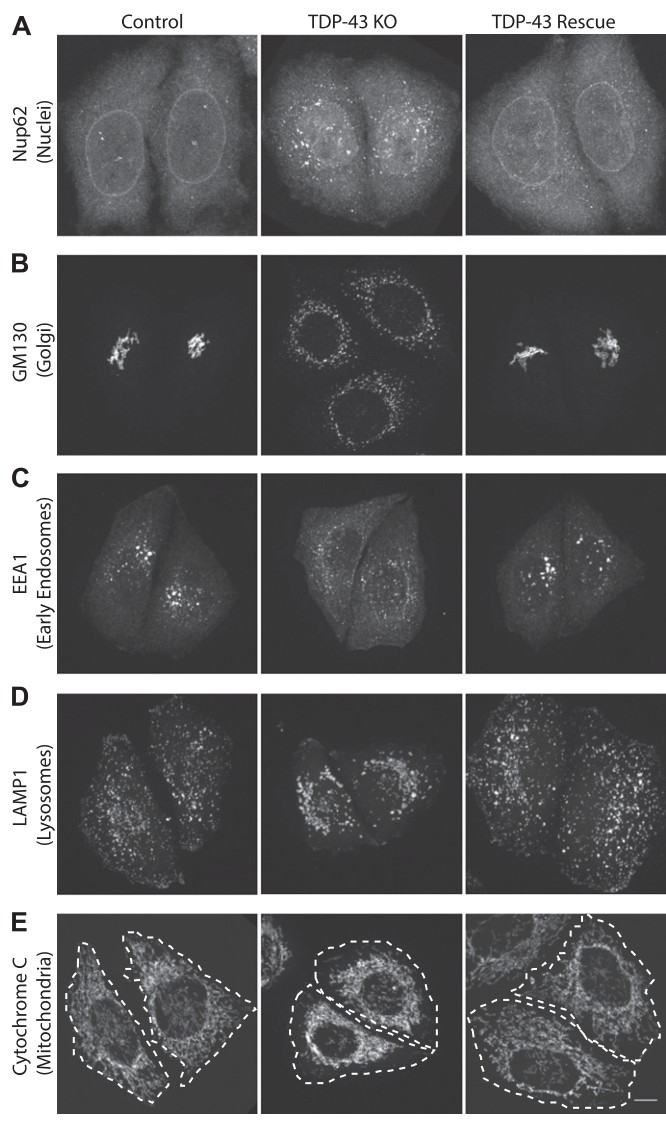

**Figure 2. TDP-43 KO affects the morphology of multiple organelles.**
**(A, B, C, D, E)** Representative confocal immunofluorescence images with nuclear pore (Nup62, panels [A]), Golgi (GM130, panels [B]), early endosome (EEA1, panels [C]), late endosome/lysosome (LAMP1, panels [D]), and mitochondria (cytochrome c, panels [E]) antibodies from cells of the indicated genotypes. Representative images are presented from the analysis of 2–3 biological replicates for each condition. Scale bar = 10 $\mu$m.

(Zhou et al, 2015; Nicholson et al, 2016), also exhibited more secretion in the TDP-43 KO cells (Fig S4).

Lysosome impairment arising from loss of TDP-43 is of potential interest in the context of neurodegenerative diseases such as ALS-FTD, where human genetics and pathology have established roles for both TDP-43 and lysosomes as contributors to the disease process. This is particularly relevant in the case of GRN-linked FTD where GRN haploinsufficiency results in neurodegeneration that is accompanied by TDP-43 cytoplasmic aggregation (Mackenzie, 2007; Rademakers et al, 2012; Gotzl et al, 2016). Our observations of lysosome defects arising from TDP-43 depletion further suggest the possibility of a toxic feed forward loop wherein initially small losses of either TDP-43 or lysosome defects such as those arising from reduced progranulin levels could be amplified.

## TDP-43 is not essential for the formation of arsenite-induced stress granules

In addition to its splicing functions within the nucleus, TDP-43 is also found in cytoplasmic stress granules and has been implicated in their formation (Freibaum et al, 2010; Liu-Yesucevitz et al, 2010; Aulas et al, 2012). We observed that arsenite-induced stress granules containing T-cell–restricted intracellular antigen-1 (TIA1, an RNA-binding protein) still formed in the absence of TDP-43 (Fig S5). Although we cannot rule out subtle defects in the kinetics of stress granule formation or effects on the formation of stress granules in response to other stimuli, this result indicates that TDP-43 is not absolutely required for stress granule formation.

## Transcriptomic analysis of TDP-43 KO cells

TDP-43 is well characterized for its ability to regulate mRNA splicing (Barmada, 2015; Ling et al, 2015; Gao et al, 2017). To gain a global view of mRNA changes arising from the absence of TDP-43, we performed RNA-Seq experiments wherein we compared the transcriptome of TDP-43 KO cells with those rescued via expression of WT TDP-43. The use of the rescued cells as our reference was supported by restoration of normal TDP-43 protein levels as well as the rescue of *POLDIP3* splicing in such cells (Fig 1). Furthermore, because they were derived from the same KO clonal population, they control for any other adaptations or mutations that might have arisen in the course of generating the TDP-43 KO. Transcriptomic analysis of these two cell populations revealed numerous genes whose expression levels were significantly altered in response to the presence or absence of TDP-43 (Fig 4A and Table S1). Featured prominently amongst the large group of transcripts whose levels were decreased in the TDP-43 KO cells are established TDP-43 targets (Ling et al, 2015) such as *ATG4B*, *INSR*, *PFKP*, and *RANBP1*. These changes in gene expression could arise because of direct effects of TDP-43 on the splicing of the affected mRNAs. For example, because of the inclusion/exclusion of exons resulting in frameshift mutations and degradation by nonsense mediated decay. However, indirect compensatory responses to the KO likely also contribute to the numerous more subtle changes that occurred in cells lacking TDP-43. We, therefore, next sought to more specifically identify genes whose splicing was altered in the TDP-43 KO cells (Fig 4B and Table S2).

was restored to normal levels in TDP-43 rescue cells (Fig 3D and F). Based on recent reports of efficient progranulin processing into granulin peptides within lysosomes (Holler et al, 2017), the increased abundance of full-length progranulin is consistent with a defect in its lysosome delivery. Impaired function of cathepsin L, a major protease that mediates conversion of progranulin into granulins (Holler et al, 2017; Lee et al, 2017), could also contribute to this phenotype. In parallel with defects in lysosome localization and processing, we also observed greater secretion of the unprocessed forms of lysosomal proteins, including progranulin and cathepsins B, D, and L (Fig S4). Interestingly, levels of prosaposin, a protein that interacts with PGRN and directs its trafficking to lysosomes

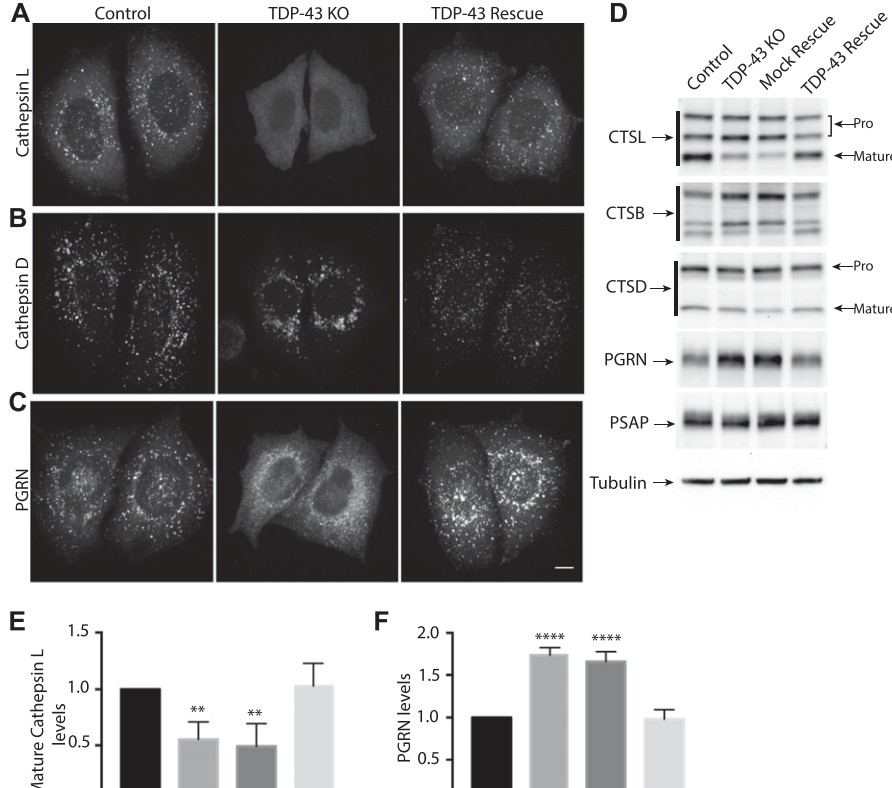

**Figure 3. Characterization of lysosome-related defects in TDP-43 KO cells.**
**(A)** Representative confocal images of Cathepsin L localization in WT, TDP-43 KO, and rescued HeLa cell lines, respectively. **(B)** Cathepsin D localization in cells of the indicated genotypes. **(C)** Impact of TDP-43 on PGRN subcellular localization. Representative images are presented from three biological replicates for experiments that are summarized in panels (A, B, and C). Scale bar = 10 $\mu M$. **(D)** Immunoblot analysis of lysosomal protein abundance and processing in cells of the indicated genotypes. Mock rescue refers to TDP-43 KO cells stably expressing empty vector. **(E)** Quantification of the TDP-43–dependent changes in the abundance of the mature form of cathepsin L (n = 4 biological replicates, $P < 0.005$, ANOVA with Bartlett posttest). **(F)** Quantification of TDP-43–dependent changes in PGRN protein levels (n = 4 biological replicates, $**P < 0.0001$, ANOVA with Bartlett posttest). Error bars show mean ± SEM.

## Identification of Nup188 as a novel TDP-43 target

Amongst the list of genes whose expression changed dramatically in a TDP-43–dependent manner was nucleoporin 188 (*Nup188*), whose levels fell by ~6 fold in the absence of TDP-43 (Fig 4F and Table S1). Analysis of differences in isoform abundance between the KO and rescued cell lines revealed that this change in *Nup188* expression in the TDP-43 KO cells was accompanied by the insertion of a cryptic 34-bp exon (Fig 4C). In contrast, transcripts containing this cryptic exon are essentially undetectable in the TDP-43–expressing cells (Fig 4C). The insertion of this 34-nucleotide cryptic exon results in a frameshift and the presence of a premature stop codon in exon 4 of *Nup188* (out of 44 in total). This is, therefore, predicted to result in nonsense-mediated decay of this transcript and is consistent with the low levels of this *Nup188* transcript in the TDP-43 KO cells.

Closer analysis of the intronic sequence surrounding the cryptic exon in *Nup188* predicts a direct role for TDP-43 in suppressing the inclusion of this exon. The TDP-43 RRMs bind preferentially to RNA sequences containing tandem UG repeats and a hallmark of TDP-43–regulated splicing events is the presence of a UG-rich tract located 3′ of cryptic exons whose splicing is suppressed by TDP-43 (Tollervey et al, 2011; Xiao et al, 2011; Lukavsky et al, 2013; Ling et al, 2015). Analysis of this region in *Nup188* reveals 25 UG pairs in the 60 base pairs immediately downstream of the 3′ splice site (Fig 4D).

The presence of this robust consensus TDP-43–binding site strongly suggests that *Nup188* splicing is directly controlled by a canonical TDP-43–mediated mechanism. Further analysis of sequencing reads that span exon boundaries within this region clearly demonstrated the preferential inclusion of the cryptic exon in the TDP-43 KO cells and its complete absence when TDP-43 is present (Fig 4E). This was further confirmed by RT-PCR analysis of amplicons spanning from exon 3 to exon 4 (Fig 4F). Immunoblot experiments also detected a major reduction in Nup188 protein levels in the TDP-43 KO cells (Fig 4G) that was rescued by re-expressing TDP-43. This major dependence of *Nup188* splicing on TDP-43 was independently replicated after CRISPR-mediated depletion of TDP-43 in HEK293FT cells (Fig 4H).

As Nup188 plays an important role as a scaffold that contributes to nuclear pore assembly (Andersen et al, 2013), the reduction in Nup188 protein levels in TDP-43 KO cells may contribute to the abnormal nuclear pore staining that we observed in these cells (Fig 2A). However, dysregulation of other previously reported TDP-43 targets implicated in nuclear pore function such as *RANBP1* are also likely to also contribute to this phenotype (Ling et al, 2015; Stalekar et al, 2015). Indeed, *RANBP1* levels were also decreased in the TDP-43 KO cells in our transcriptomic analysis, although to a more modest degree than what was observed for *Nup188* (Fig 4A and Table S1). In spite of these changes, the viability of the TDP-43 KO cells argues strongly for intact compartmentalization of the nucleus.

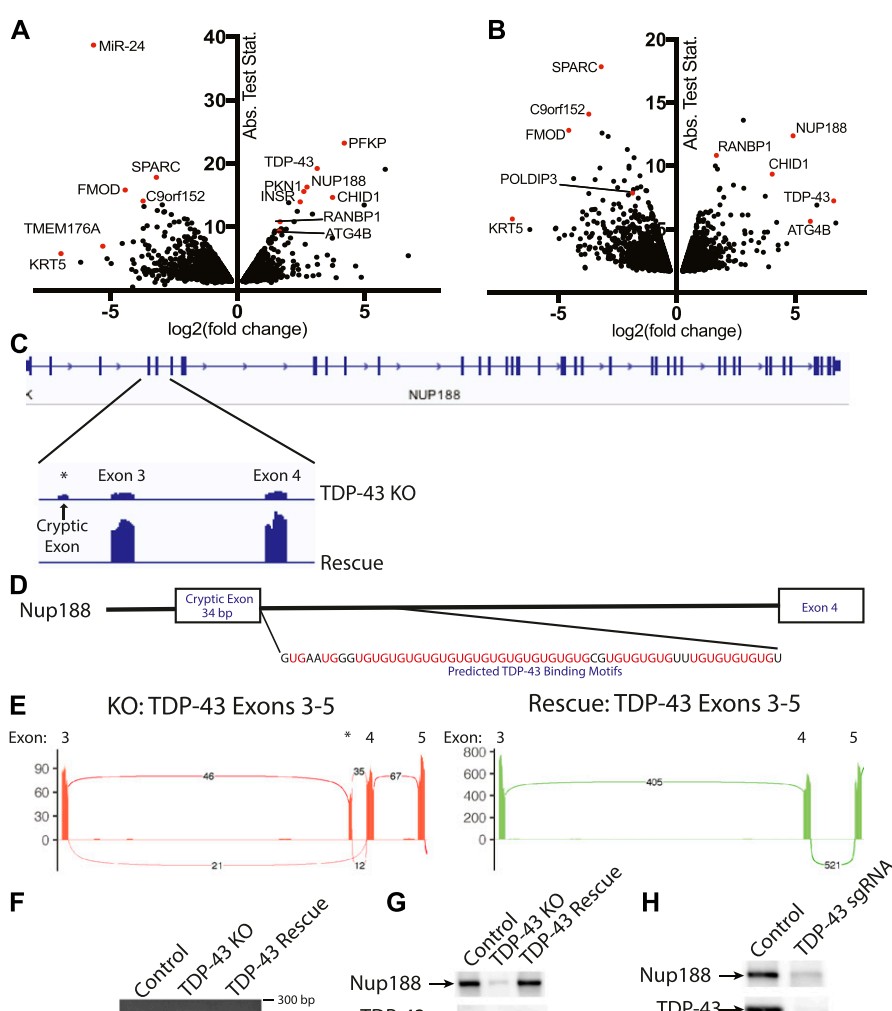

**Figure 4. Identification of a cryptic exon in Nup188 as a TDP-43 target.**
**(A)** Changes in overall transcript abundance (per gene) revealed by analysis of RNA-Seq data from TDP-43 KO versus rescued cell lines. Fold change calculated for rescue/KO HeLa cells. Rescue indicates TDP-43 KO cell line stably expressing exogenous TDP-43. **(B)** Changes in individual transcript abundance detected by analysis of RNA-Seq data from TDP-43 KO versus rescued cell lines. Fold change calculated for rescue/KO cells. Three biological replicates were analyzed in these RNA-Seq experiments. The change in the TDP-43 transcript reflects the use of an exogenous TDP-43 transgene, which lacks 5′ and -3′ UTRs for the rescue cell line. **(C)** Alignment of RNA-Seq reads showing the reduced overall abundance of *Nup188* transcripts in TDP-43 KO cells as well as the increased abundance of a cryptic exon. **(D)** Presence of UG tandem repeats in the intron following the *Nup188* cryptic exon. **(E)** Sashimi plots summarizing splicing patterns in the exon 3 to exon 5 region of TDP-43 in KO and rescue cell lines, respectively. **(F)** RT-PCR validation of altered *Nup188* splicing in TDP-43 KO cells. **(G)** Immunoblot analysis reveals the impact of TDP-43 KO on Nup188 protein levels. **(H)** Immunoblot analysis of Nup188 levels in Control versus TDP-43 depleted HEK293FT cells.

To examine this problem, we made use of a NLS-td-Tomato nucleocytoplasmic shuttling reporter that was previously used to investigate ALS disease mechanisms (Zhang et al, 2015). As expected, based on the previous use of this reporter (Zhang et al, 2015), the signal mainly resided in the nucleus in both control and TDP-43 KO cells, and quantitative analysis of the nuclear/cytoplasmic fluorescence ratio did not show any differences between the respective genotypes (Fig S6A and B). These observations raise questions about selective pressures and posttranscriptional mechanisms that may have allowed the TDP-43 KO cells to adapt to survive in the absence of TDP-43.

## Cation-independent mannose-6-phosphate receptor (CI-M6PR) levels are reduced in TDP-43 KO cells

The impaired lysosomal delivery of multiple luminal proteins in TDP-43 KO cells raised questions about underlying mechanisms. Although many factors likely contribute to these lysosome-related phenotypes (including alterations that we observed in Golgi, endosome and lysosome morphology [Fig 2B]), we noted a ~2× reduction in the abundance of *CI-M6PR* (also known as IGF2R)

expression in TDP-43 KO cells (Table S1). CI-M6PR is a well-characterized sorting receptor for the trans-Golgi network to endosome trafficking of multiple lysosomal hydrolases (Ghosh et al, 2003). Two factors motivated us to explore this change in more detail. First, to test whether even modest expression changes identified by RNA-Seq were indicative of detectable changes at the protein level. Second, to gain further insight into cellular changes that could contribute to the lysosome defects in TDP-43 KO cells. Consistent with the RNA-Seq results, we observed a parallel ~2× decrease in the overall abundance of the CI-M6PR protein in the TDP-43 KO cells (Fig S7A and B). In contrast to the CI-M6PR, sortilin (SORT1), another sorting receptor involved in the trafficking of lysosomal hydrolases, did not show changes in transcript abundance, splicing or protein levels (Tables S1 and S2; Fig S7A) even though its splicing was previously reported to be regulated by TDP-43 (Prudencio et al, 2012). This may reflect known species-specific preferences for the mouse *SORT1* versus human *SORT1* transcripts in the requirement for TDP-43 in regulating their splicing and/or cell type–specific differences in this function of TDP-43 (Prudencio et al, 2012; Mohagheghi et al, 2016).

### Disease-causing TDP-43 mutants rescue KO phenotypes but also reveal defects specific to individual mutations

Despite intense interest in elucidating the mechanisms whereby TDP-43 mutations give rise to neurodegenerative disease, it still remains unclear to what extent these mutations exert pathogenic effects via gain-of-function versus loss-of-function mechanisms (Kabashi et al, 2010; Vanden Broeck et al, 2015; Orru et al, 2016). Testing the functionality of disease-causing TDP-43 mutants requires an experimentally tractable model system with loss-of-function phenotypes that can be rescued by re-expression of the WT TDP-43 at physiological levels. We, thus, took advantage of

robust TDP-43 KO phenotypes to assess the functionality of a panel of neurodegenerative disease-causing TDP-43 mutants (Fig 5A). To this end, we generated TDP-43 KO cell lines that stably expressed eight different disease-causing TDP-43 mutations (Fig 5A; A90V, P112H, D169G, K263E, A315T, Q331K, M337V, and A382T) (Banks et al, 2008; Gitcho et al, 2008; Kabashi et al, 2008, 2010; Rutherford et al, 2008; Sreedharan et al, 2008; Winton et al, 2008; Corrado et al, 2009; Kovacs et al, 2009; Moreno et al, 2015; Vanden Broeck et al, 2015). After confirming that each mutant was expressed at similar levels to the WT TDP-43 (Fig 5B), we next examined their ability to rescue the expression levels and/or splicing of POLDIP3, PFKP (a previously characterized TDP-43 target gene whose splicing is altered in the

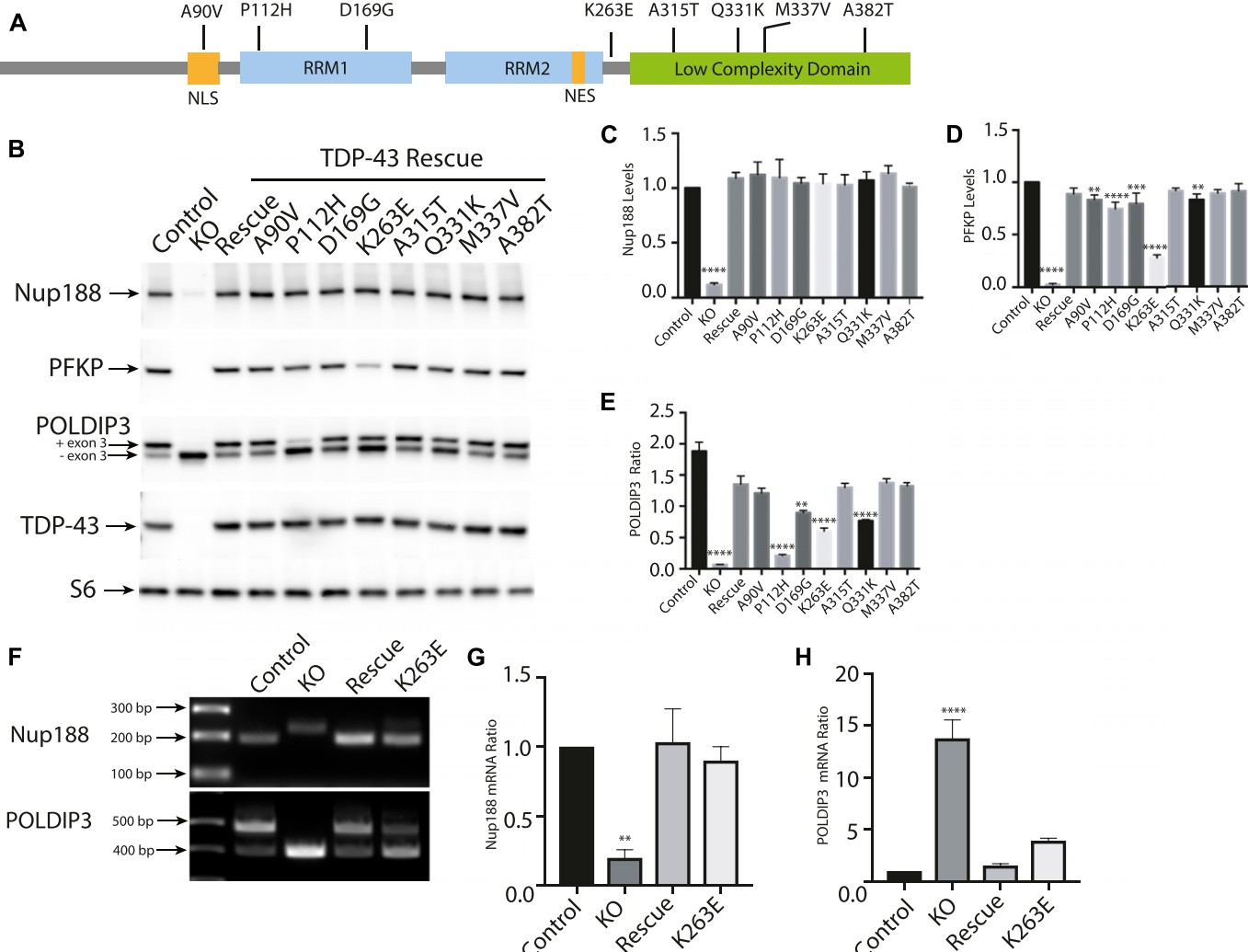

**Figure 5. ALS-linked TDP-43 mutations selectively rescue the regulation of specific TDP-43 targets.**
**(A)** Schematic diagram that summarizes the TDP-43 protein domain organization and the location of mutants that were investigated in this study. **(B)** Representative immunoblots from control, TDP-43 KO, and TDP-43 KO HeLa cell lines that were reconstituted with TDP-43 variants containing the indicated ALS causing mutations. **(C)** Quantification of the ability of mutant forms of TDP-43 to rescue Nup188 abundance (n = 3 biological replicates; ****$P < 0.0001$; ANOVA with Dunnett's multiple comparison test, comparison with WT). **(D)** Quantification of PFKP abundance (n = 3 biological replicates; ****$P < 0.0001$, ***$P < 0.0006$, and **$P < 0.0054$; ANOVA with Dunnett's multiple comparison test to compare mutant cell lines with WT). **(E)** Ratio of the POLDIP3 exon 3 inclusion (n = 3 biological replicates; ****$P < 0.0001$, ***$P < 0.0003$, and **$P < 0.0016$; ANOVA with Dunnett's multiple comparison test to test for differences between the mutant cell lines versus WT rescue samples). Error bars show mean ± SEM. **(F)** RT-PCR showing effects of rescue with WT, TDP-43, and K263E mutant on Nup188 and POLDIP3 splicing. **(G, H)** Quantification of RT-PCR results for Nup188 and POLDIP3, respectively. In both cases, the density of the lower bands were measured using ImageJ and divided by the density of the upper bands. Results were normalized to control (n = 3 biological replicates; **$P < 0.0067$ and ****$P < 0.0001$; ANOVA with Dunnett's multiple comparison test). Error bars show mean ± SEM.

elderly human brain [Ling et al, 2015; Raj et al, 2018]), and *NUP188*. Interestingly, even though many combinations of TDP-43 mutant and target gene displayed full rescue of the KO phenotype, there were also multiple cases where the rescue did not occur (Fig 5B–D). For example, multiple mutants that rescued *Nup188* and *PFKP* phenotypes failed to fully rescue the splicing of *POLDIP3*. These observations could at least partially reflect the fact that the suppression of cryptic exon inclusion mechanism whereby TDP-43 regulates *Nup188* and *PFKP* is distinct from promoting exon retention in POLDIP3 (Fiesel et al, 2012; Ling et al, 2015). Consistent with the previous identification of an important role for RRM1 in supporting the regulation of *POLDIP3* (also known as *SKAR*) splicing by TDP-43 (Fiesel et al, 2012), we observed that the P112H mutant rescued Nup188 and PFKP phenotypes but was deficient in restoring normal splicing of *POLDIP3* (Fig 5B–E). However, additional factors must determine the functionality of specific TDP-43 mutant proteins as we found that the P112H mutant fully supported the regulation of Nup188 and PFKP but not POLDIP3. Meanwhile, the K263E mutant rescued Nup188 protein levels completely but only partially supported regulation of PFKP and POLDIP3 protein levels and isoform ratios, respectively (Fig 5B–E). This was furthermore confirmed at the mRNA level where RT-PCR analysis revealed that K263E mutant fully supported *Nup188* cryptic exon exclusion (Fig 5F and G) while only partially restoring *POLDIP3* splicing (Fig 5F and H). Collectively, these results indicate that in addition to loss of nuclear functions for TDP-43 mutants as they aggregate in the cytoplasm, individual TDP-43 mutants are also uniquely compromised in their ability to support the splicing of specific transcripts. These results stand in contrast to a previous report that disease-causing mutations have dominant effects on splicing (Arnold et al, 2013). This difference in conclusions could reflect the examination of physiological levels of mutant TDP-43 proteins in a KO background in our study versus over-expression on top of WT TDP-43 protein by Arnold et al. Although the panel of mutations that we examined did not reveal striking patterns between the regions where mutations resided and phenotypic consequences, the identification of differences in the functionality of these mutant proteins opens the door to future studies that would investigate in more detail whether mutation-specific defects in splicing ability have any specific impact on variables of disease pathology such as the propensity to cause ALS versus FTD, age of onset, and disease severity. Likewise, from a cell biology and biochemistry perspective, the distinct functionality profiles of different TDP-43 mutants could serve as valuable tools for dissecting mechanisms of action. As the phase separation properties of RNA-binding proteins have recently become widely appreciated (Guo et al, 2019), the possibility that differences in functionality that we have observed may relate to such characteristics is a worthy topic for future studies that would shed light on TDP-43 functions in both health and disease. More generally, our results establish the utility of this human TDP-43 KO cell line as a valuable tool for the investigation of these and other disease-causing TDP-43 mutations in the future.

### Summary

Our focus on the essential functions of TDP-43 in HeLa cells was driven by the practical issues relating to viability that have

precluded the complete KO of TDP-43 in other model systems. Although this strategy provides insights into the role for TDP-43 in the regulation of cellular processes that are common to all cell types, there are likely to be additional targets of TDP-43 that are unique to specialized cell types such as neurons, muscles, and glia. Indeed, the pattern of splicing changes arising from TDP-43 depletion has been reported to vary between cell types (Jeong et al, 2017). Nonetheless, by taking advantage of a cell culture model that uniquely offered the opportunity to completely eliminate TDP-43 function, we have generated multiple new insights that broaden the understanding of the pleiotropic cell biological consequences of eliminating TDP-43 expression and shed light on the functionality of disease-causing TDP-43 mutants. Having observed that loss of TDP-43 affects multiple cellular organelles and hundreds of mRNA transcripts, it is challenging to attribute putative disease-causing mechanisms to the altered regulation of any one single transcript. However, because of the tremendous interest currently focused on defects in nuclear pore function in neurodegenerative disease associated with TDP-43 pathology (Ward et al, 2014; Zhang et al, 2015, 2018; Kim & Taylor, 2017; Chou et al, 2018), it is interesting to consider how the Nup188 defects that we have uncovered in TDP-43 KO cells may intersect with other previously identified TDP-43–dependent components of the nucleocytoplasmic transport machinery such as RANBP1 (Ling et al, 2015; Stalekar et al, 2015). Our analysis of the ability of mutant forms of TDP-43 to rescue the splicing and expression levels of target genes in TDP-43 KO cells established that these cells are a useful tool for investigating the functionality of disease-causing TDP-43 mutations. Our data furthermore raise new questions about whether subtle differences in TDP-43 protein functionality contributes to the pathology arising from distinct TDP-43 mutations.

## Materials and Methods

### Cell culture

HeLaM cells were kindly provided by Pietro De Camilli (Yale University) and were grown in DMEM, 10% FBS, and 1% penicillin/streptomycin supplement (all from Thermo Fisher Scientific). 293FT cells (Thermo Fisher Scientific) were grown on dishes coated with 0.1 mg/ml poly-d-lysine (Sigma-Aldrich) in the media described above. For stress granule induction, the cells were incubated with 0.5 mM sodium arsenite (Fluka Analytical) before fixation.

### CRISPR/Cas9 genome editing

Use of CRISPR/Cas9 genome editing to generate KO cell lines was previously described (Amick et al, 2016). The guide RNA sequences used to generate TDP-43 KO cell lines (summarized in Table S3) were annealed and cloned into the BbsI site of pSpCas9(BB)-2A-Puro vector (px459, Feng Zhang; Addgene). Sub-confluent HeLa cells were transfected with 1 μg of px459 vector using FuGENE 6 (Promega). Transfected cells were selected with 2 μg/ml puromycin for 2 d and surviving cells were subsequently plated in 96-well plates at 1 cell/well density to generate clonal lines. After selection

and expansion of clonal populations, KOs were first identified by Western blotting and subsequently confirmed by sequencing of PCR-amplified genomic DNA (primers summarized in Table S3). For HEK293FT experiments, a polyclonal TDP-43–depleted population was examined after puromycin selection but without clonal selection.

### Lentiviral transduction

For rescue experiments, the human *TDP-43* cDNA sequence was PCR-amplified from a previously described TDP-43–tdTomato expression vector (kindly provided by Dr. Zuoshang Xu via Addgene [Yang et al, 2010], see Table S3 for primer sequences). The PCR product was introduced into the pLVX-puro vector (Clontech) by Gibson assembly (NEB). For ALS-associated TDP-43 mutations, cDNAs containing the mutations of interest were synthesized (gBlocks; IDT) and introduced into the pLVX-puro vector by Gibson Assembly (these new plasmids will be made publicly available through Addgene). For lentivirus production, 293FT cells (Invitrogen) were co-transfected with pLVX-TDP-43 along with the lentiviral packaging vectors pCMV-VSV-G (#8454; Addgene) and pSPAX2 (#12260; Addgene) in equal amounts using FuGENE 6 transfection reagent. The lentivirus-containing media was collected 24 h posttransfection, filtered through a 0.45-$\mu$m filter, combined with 8 $\mu$g/ml polybrene and added to the TDP-43 KO cell line. Transduced cells were selected with 2 $\mu$g/ml puromycin to establish polyclonal stable lines.

### Immunofluorescence, microscopy, and image analysis

Cells grown on 12-mm No. 1.5 coverslips (Carolina Biological Supply) were fixed with 4% paraformaldehyde (Electron Microscopy Sciences), washed with PBS, and permeabilized with either 0.1% TX100 or 0.1% saponin in a 3% BSA/PBS buffer. Subsequent primary and secondary antibody incubations were carried out in the permeabilization buffer. Coverslips were mounted in ProLong Gold reagent supplemented with DAPI (Invitrogen). Images were acquired with either a Zeiss LSM 800 laser scanning confocal microscope with a 63× Plan Apo (NA = 1.4) oil immersion objective and Zeiss Efficient Navigation software or an UltraVIEW VoX spinning disk confocal microscope (PerkinElmer) that consisted of a Nikon Ti-E Eclipse inverted microscope equipped with 60× CFI PlanApo VC, NA 1.4, oil immersion objective and a CSU-X1 (Yokogawa) scan head that was driven by Volocity (PerkinElmer) software. For live cell imaging analysis of microtubules and F-actin, the cells were labeled with SiR-tubulin and SiR-actin (Cytoskeleton Inc.) as per the manufacturer's instructions. An automated analysis pipeline was developed with Cell Profiler (McQuin et al, 2018) for the quantification of nuclear versus cytoplasmic ratios of the NLS-td-Tomato-NES reporter (Zhang et al, 2015).

### Immunoblotting

Cells were lysed in TBS supplemented with 1% Triton X-100 and protease/phosphatase inhibitor cocktails (Roche Diagnostics), the insoluble material was cleared by centrifugation for 10 min at 20,000*g*. Immunoblotting was performed by standard methods using 4–15% Mini-PROTEAN TGX precast polyacrylamide gels and nitrocellulose membranes (Bio-Rad). Ponceau S staining of membranes was routinely used to assess equal sample loading and transfer efficiency. Blocking was performed in 5% milk in TBS with 0.1% Tween 20. Primary antibody incubations were performed in 5% BSA in TBS with 0.1% Tween 20. Signals were detected with HRP-conjugated secondary antibodies (Cell Signaling Technology) and either Super Signal West Pico or Femto chemiluminescent detection reagents (Thermo Fisher Scientific) on a VersaDoc imaging system (Bio-Rad). ImageJ was used to measure band intensities. Antibodies used in this study are listed in Table S4.

### RNA isolation and RT-PCR

Total RNA was extracted from WT, TDP-43 KO, and rescue cell lines using the RNeasy kit (QIAGEN). 400 ng of total RNA was reverse-transcribed using the iScript cDNA synthesis kit (Bio-Rad). Cryptic and alternatively spliced exons were PCR-amplified from the resulting cDNA with Q5 polymerase (NEB) and either Nup188 or POLDIP3 primers (IDT). The primers used to amplify the cryptic *Nup188* exon were as follows: 5′-GGAGCAGTAGAGAACTGTGG-3′ and 5′-GCTGATTCTTAAACCCAGTTC-3′. The primers to amplify *POLDIP3* splice variant were: 5′-GCTTAATGCCAGACCGGGAGTTG-3′ and 5′-TCATCTTCATCCAGGTCATATAAATT-3′. The resulting PCR reactions were resolved on 2.5% agarose gels supplemented with SYBR green (Invitrogen) and visualized on a VersaDoc imaging system (Bio-Rad).

### RNA-Seq

Six samples (three pairs of biological replicates of TDP-43 KO and rescue) were sequenced on Illumina's HiSeq 2500 using 2 × 70-bp paired-end reads, generating 104.4–128.2 million reads (52.2–64.1 million pairs) per sample. The sequencing reads were first trimmed for quality, trimming to the last base with quality ≥20 and dropping any read pair where either of the trimmed reads was shorter than 45 bp. The trimmed reads were aligned to the hg19 reference genome using TopHat2 (Trapnell et al, 2009), an RNA-Seq aware aligner that allows for cross-exon, split alignments of reads to the genome. Overall, 96.8–97.1% of the paired-end reads aligned to the human genome, with an ~94.2–94.8% concordant pair alignment rate.

The Cufflinks and Cuffmerge software (Trapnell et al, 2013) was used to assemble the transcript isoforms from each sample's reads and to merge the per-sample transcripts with known human transcripts to form a set of known and novel transcript isoforms from the data. This computation involves grouping overlapping reads together into "bundles," constructing an "overlap graph" representing contiguously aligned regions and linked regions (where spliced alignments in a bundle link the aligned regions), then identifying the most parsimonious set of transcript isoforms that cover the overlap graph.

Transcript abundance calculation and differential gene expression analysis, using the assembled known plus novel transcripts, was performed using Cuffdiff (Trapnell et al, 2013). Heat maps and charts were visualized using Cummerbund (Trapnell et al, 2012), and Integrative Genomics Viewer (Thorvaldsdottir et al, 2013)

was used for visualization of read depths and alignments. Sashimi plots were generated with ggsashimi using the alignment files generated with TopHat2 and hg19 annotation (Garrido-Martin et al, 2018). The RNA-Seq data generated in this study is available via GEO accession number GSE136366.

## Supplementary Information

## Acknowledgements

We are grateful to Patrick Lusk for reagents and advice related to analysis of the nuclear envelope. We appreciate support from the Yale Center for Genome Analysis for RNA-Seq experiments and from Sameet Mehta and James Knight for data analysis related to these experiments. Grants from the National Institutes of Health (GM105718, AG062210, and AG047270), and The Bluefield Project to SM Ferguson provided financial support for this research.

### Author Contributions

A Roczniak-Ferguson: conceptualization, data curation, formal analysis, validation, investigation, visualization, methodology, and writing—original draft, review, and editing.
SM Ferguson: conceptualization, formal analysis, supervision, funding acquisition, visualization, project administration, and writing—original draft, review, and editing.

### Conflict of Interest Statement

The authors declare that they have no conflict of interest.

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
