## [Reviewer comments · Life Science Alliance]

Life Science Alliance

Pleiotropic requirements for human TDP-43 in the regulation of cell and organelle homeostasis

Shawn Ferguson and Agnes Roczniak-Ferguson

DOI: <https://doi.org/10.26508/lsa.201900358>

Corresponding author(s): Shawn Ferguson, Yale School of Medicine

Review Timeline:	Submission Date:	2019-02-24
	Editorial Decision:	2019-03-21
	Revision Received:	2019-08-11
	Editorial Decision:	2019-08-15
	Revision Received:	2019-08-26
	Accepted:	2019-09-02

Scientific Editor: Andrea Leibfried

Transaction Report:

March 21, 2019

Re: Life Science Alliance manuscript #LSA-2019-00358

Dr. Shawn Michael Ferguson
Yale School of Medicine
Department of Cell Biology
295 Congress Ave BCMM254E
New Haven, CT 06510

Dear Dr. Ferguson,

Thank you for submitting your manuscript entitled "Pleiotropic requirements for human TDP-43 in the regulation of cell and organelle homeostasis" to Life Science Alliance. The manuscript was assessed by expert reviewers, whose comments are appended to this letter.

As you will see, your work received split views from the reviewers, with reviewer #1 seeing little value in analyzing loss of TDP-43 in HeLa cells. All reviewers furthermore note technical issues such as insufficient controls for successful knock-out, lack of sufficient numbers of replicates and rescue experiments as well as the need for further support as well as clarifications on some parts. We discussed your work in light of these comments and decided to invite you to submit a revised version of your manuscript for publication here. Importantly, all technical issues need to get addressed. Reviewer #1 and #3 furthermore provide constructive input on how to increase the value provided by your findings by slightly extending the analysis, and we would like to encourage you to follow their suggestions.

Thank you for this interesting contribution to Life Science Alliance. We are looking forward to

receiving your revised manuscript.

Sincerely,

B. MANUSCRIPT ORGANIZATION AND FORMATTING:

Reviewer #1 (Comments to the Authors (Required)):

In the manuscript "Pleiotropic requirements for human TDP-43 in the regulation of cell and organelle homeostasis", the authors have used Cas9/CRISPR technology to generate a complete TDP-43 knockout in the HeLa cell line for transcriptomics and immunocytochemistry analysis. 1) Loss of TDP-43 led to morphological/localization defects in the nuclear envelope, Golgi, endosomes, lysosomes, and mitochondria. 2) Loss of TDP-43 led to Nup188 splicing alterations. 3) Rescue experiments with multiple ALS-associated TDP-43 mutant variants demonstrated a broad ability to support splicing of several but not all tested TDP-43 target transcripts.

There is considerable interest in the role of TDP-43 in the disease process of ALS and other neurodegenerative diseases. TDP-43 proteinopathy is characterized by loss of TDP-43 from the nucleus and the accumulation of hyperphosphorylated and ubiquitinated TDP-43 fragments in mainly cytoplasmic aggregates. It is likely that both loss of nuclear TDP-43 and gain of toxic function of aggregates contribute to the disease, and studies focused on loss of function phenotypes have merit. While some of the observations in this manuscript are of potential interest, at this stage the findings are of limited value for the community, as outlined in more detail below.

1) Various studies have used shRNA constructs in neuronal cell lines and primary neurons, as well as KO in animals to investigate the role of TDP-43. It is not clear why HeLa cells should be a relevant disease model. Neuronal cells appear to be more sensitive to TDP-43 pathology than cancer cells.

2) The authors speculate that the frameshift will lead to NMD, but loss of the N-terminal protein fragment is not shown. Only one antibody is used to demonstrate loss of TDP-43 protein after KO. This antibody was raised against a peptide around aa260 right after RRM2 and will not recognize truncated proteins. This is important, since also the N-terminal part may have toxic effects. Polyclonal antibodies raised against an N-terminal fragment of recombinant TDP-43 are available from PTG.

3) For many experiments it is not clear how many times they have been repeated independent experiments (unless they are quantified).

4) Do morphological defects of organelles lead to functional deficits? E.g. does aberrant Nup62 staining lead to transport defects? This would be important to investigate.

5) Are morphological and localization defects of organelles caused by defects in the MT or actin cytoskeleton?

6) Can expression of Nup188 rescue morphological and potential functional defects?

7) Rescue experiments with mutant TDP-43 are of limited value. It is unlikely that as the authors state, "as some TDP-43 disease causing mutants failed to support the regulation of specific target transcripts, our results raise the possibility of mutation-specific loss-of-function contributions to disease pathology." This would be of more interest if nuclear localization of TDP-43 were preserved in these patients. However, mutant TDP-43 in fALS still leads to loss of TDP-43 from the nucleus and cytoplasmic aggregation.

Reviewer #2 (Comments to the Authors (Required)):

In this work, Roczniaak-Ferguson and Ferguson SM have analyzed the physiological role of TDP-43

in multiple cell lines. In particular, they have focused their attention on splicing regulation of Nup188 and on the role played by TDP-43 disease-associated mutations. Overall, this study has been technically well performed. Although many of the conclusions have been conformed by previous studies there are also some interesting observations with regards to the types of proteins that are regulated by TDP-43 and the role played by mutations. A few clarifications and additions are nonetheless in order:

- 1) What are the splicing changes detected in the TDP-43 transcript that are highlighted in Fig.4B? The authors might also want to describe a little more this experiment by mentioning how many of these events belong to skipped exons, cryptic splice site activation, intron retention, etc.
- 2) Additional information should be provided with regards to the 34 nt cryptic exon inserted in the Nup188 transcript following TDP-43 deletion. Is this exon predicted to introduce the presence of a premature stop codon in the mature mRNA sequence?. If so, then the drop in transcript expression could be due to activation of the NMD pathway. Has this been tested by treating the cells with an MND inhibitor?.
- 3) Authors should note that the difference between the splicing profile of mouse vs. human SORT1 was explained by the presence of differential splicing regulators in its exon 17b (Mohagheghi F et al., 2016).
- 4) It is rather strange that mutations which impair splicing efficiency (ie. K263E) can rescue Nup188 expression that, according to the results of the author, derives from the inclusion of a cryptic exon. It would be very interesting to know whether in the cells that stably express the K263E mutant the cryptic exon is not excluded any more. Has this been tested?. In general, at least for the two examples that deal with splicing regulation (Nup188 and POLDIP3) the authors should also add a quantification of the different splicing isoforms in addition to the Western blots.

Reviewer #3 (Comments to the Authors (Required)):

Loss of function of TDP43 is now recognized as a key driver of neurodegeneration in FTD and ALS, but the underlying mechanisms remain unclear. In this manuscript the authors describe the creation and characterization of a novel TDP-43 knockout human cell line, the first detailed characterization of how loss of TDP-43 affects human cells. They then characterize general organelle morphology, focusing on lysosomal properties, and perform transcriptomics comparing the TDP-43 KO line and a rescued control. They identify a novel splice target of TDP-43, Nup188, and then demonstrate that while TDP-43 is needed for splicing activity, several distinct TDP-43 disease mutations do not abolish the ability of TDP-43 to splice Nup188. However, they also show that different disease mutations can selectively alter the ability of TDP-43 to splice specific targets. The TDP-43 KO line and disease mutation lines developed in this study will be useful for future studies aimed at uncovering the role of TDP-43 gain and loss of function in disease pathology.

Overall, I feel that this manuscript address a critical gap in our knowledge of TDP43 function, and the experiments were exceptionally well-conceived and rigorously conducted. There are a few suggestions that the authors may wish to consider to make the paper even stronger.

1. This study would significantly benefit from transcriptomics of at least one or two key disease mutant lines compared to the KO studies presented. While a detailed mechanism of why TDP43 mutants fail to completely rescue KO cells is beyond the scope of this paper, it would be helpful to know to what degree mutants can rescue function using a scalable technique. Additional forward-thinking discussion regarding possible mechanisms would also be helpful. For example, we now know that mutant RNA granule proteins can alter the properties of other RNA granule proteins through dysregulated phase separation properties of multi-protein RNPs. How much of the

observed lack of rescue is due to pure loss of function of TDP-43 versus dysregulated biology of other members of an RNP due to dysregulated droplet biology?

2. It would be helpful to test what are the predominant TDP43 isoforms in HeLa cells (perhaps these data are even available on online databases). There is at least one additional major TDP-43 isoform that has been described (D'Alton, RNA, 2015; Seyfried et al Mol Cell Proteomics 2010) that lacks the NLS, and recent unpublished data presented at national meetings suggests that this may be a major isoform in vulnerable neurons. Since the rescue studies done here only utilize the full-length isoform, loss of function in neurons that express multiple isoforms may be more complicated.
3. The cathepsins are described in their relation to lysosomal localization in the TDP-43 KO line, but this data is not formally shown. For example, Figure 3B shows punctate Cathepsin D, but does not demonstrate that these structures are lysosomes. This interesting result of distinct mechanisms for Cathepsin delivery should be formally shown perhaps with colocalization of Cathepsin D and LAMP1.
4. The authors state in Figure 3C that progranulin delivery to lysosomes is diminished. While the cytoplasmic buildup of progranulin in the TDP-43 KO is clear in the image, it is less clear if localization to lysosomes is actually diminished (or if the cytosolic accumulation merely alters the differential between punctate and cytoplasmic signal). This should be tested perhaps by quantifying total fluorescence signal and/or colocalization to lysosomes.
5. The authors show that TDP-43 loss results in both a defect in Nup188 splicing and abnormal staining of Nup62, suggesting a defect in general nuclear pore organization. This result should be verified with staining of an additional nuclear pore marker to distinguish between a general defect or a specific interaction between Nup188 and Nup62 localization. Alternatively, a test of nuclear pore function could be performed to strengthen the modest RANBP1 result reported.

Minor comments:

1. On page 5 of the introduction, the authors state "These results suggest that loss-of-function arising from cytoplasmic accumulation and aggregation of mutant TDP-43 proteins rather than a direct loss of their fundamental splicing functions causes them to confer disease risk." However, the authors only test the ability of TDP-43 disease mutants to splice a few specific splice targets. Furthermore, they find that mutants can selectively lose the ability to splice specific targets. As such their data would also be consistent with TDP-43 mutants contributing to disease via a loss of splicing activity of one or more unidentified splice targets. They also do not examine cytosolic accumulation or aggregation of the mutant proteins expressed in the TDP-43 KO background to back this claim. We suggest revising the text to take a more agnostic view point.
2. For Figure 4 that authors describe an insertion of a cryptic 34 bp exon in Nup188 in the TDP-43 knockout lines. While the authors describe in the methods that the transcriptomics were done in triplicate, it would be interesting to know if the cryptic exon was identical between samples or if there was some variability in the absence of TDP-43 activity.

Reviewer #1

There is considerable interest in the role of TDP-43 in the disease process of ALS and other neurodegenerative diseases. TDP-43 proteinopathy is characterized by loss of TDP-43 from the nucleus and the accumulation of hyperphosphorylated and ubiquitinated TDP-43 fragments in mainly cytoplasmic aggregates. It is likely that both loss of nuclear TDP-43 and gain of toxic function of aggregates contribute to the disease, and studies focused on loss of function phenotypes have merit. While some of the observations in this manuscript are of potential interest, at this stage the findings are of limited value for the community, as outlined in more detail below.

We thank the reviewer for their careful consideration of our manuscript. We believe that our manuscript has value that extends beyond the role of TDP-43 in neurodegenerative disease by also providing new insights into the cell biological consequences of TDP-43 depletion. The cell line model that we have used has both strengths and weaknesses. We have taken advantage of the robust growth and experimental tractability of HeLa cells to generate new insights into TDP-43 function. To our knowledge, this represents the first successful attempt at generating human TDP-43 KO cells. We have used multiple approaches to characterize the cells and several novel insights have arisen from these efforts. We now seek to communicate our results to the broader research community so that they can be used by other investigators with an interest in TDP-43 function in health and disease.

1) Various studies have used shRNA constructs in neuronal cell lines and primary neurons, as well as KO in animals to investigate the role of TDP-43. It is not clear why HeLa cells should be a relevant disease model. Neuronal cells appear to be more sensitive to TDP-43 pathology than cancer cells.

We recognize the reasons for interest in the neuronal functions of TDP-43. However, we wish to emphasize that TDP-43 is a ubiquitously expressed gene and that our focus was largely on the normal functions of the TDP-43 protein. Our manuscript also outlines how the lethality of TDP-43 depletion has long prevented the use of complete TDP-43 KO cells for investigation of the fundamental functions of TDP-43. Having successfully established viable human TDP-43 KO cells, we took advantage of this new tool to investigate the cellular functions of TDP-43. We have also used these cells as a platform that reveals unexpected differences in the functionality of some disease causing TDP-43 mutations. We do not suggest that our TDP-43 KO cell line is a replacement for neurons as a disease model, however, we clearly demonstrate that it is useful for addressing specific questions related to TDP-43 function.

2) The authors speculate that the frameshift will lead to NMD, but loss of the N-terminal protein fragment is not shown. Only one antibody is used to demonstrate loss of TDP-43 protein after KO. This antibody was raised against a peptide around aa260 right after RRM2 and will not recognize truncated proteins. This is important, since also the N-terminal part may have toxic effects. Polyclonal antibodies raised against an N-terminal fragment of recombinant TDP-43 are available from PTG.

We appreciate the suggestion to take advantage of this excellent anti-TDP-43 antibody from PTG. We have now used it for immunoblot experiments and have found that it specifically recognizes the 43 kDa form of TDP-43 in WT cells and that this band is absent from our TDP-43 KO line. This new data is presented in Supplemental Figure 1E. This N-terminally directed antibody did not reveal the presence of truncated TDP-43 fragments. The revised manuscript thus presents evidence from 2 distinct antibodies that support the conclusion that our TDP-43 KO cell line lacks the TDP-43 protein. The conclusion that the phenotypes that we observe are due to loss of the normal functions of TDP-43 (rather

than a toxic gain-of-function from a truncated protein) is additionally supported by observations that re-expression of WT TDP-43 (at endogenous levels) rescues the multiple KO phenotypes that we report.

3) For many experiments it is not clear how many times they have been repeated independent experiments (unless they are quantified).

Figure legends have been updated to address this concern.

4) Do morphological defects of organelles lead to functional deficits? E.g. does aberrant Nup62 staining lead to transport defects? This would be important to investigate.

Our data defines changes to the morphology of multiple organelles. These changes demonstrate the existence of pleiotropic cell biological changes that arise in response to loss of TDP-43 function. These findings stand in contrast to some previous studies that have more narrowly focused on individual genes or pathways as targets of TDP-43 function. A key message of our manuscript is the occurrence of widespread consequences of TDP-43 depletion that have an impact on multiple organelles rather than the details related to any specific organelle or physiological process. With respect to the specific question related to nuclear-cytoplasmic transport, we have performed new experiments wherein we carefully quantified the subcellular distribution of a fluorescent reporter of nuclear import and export (Supplemental Figure 6A and B). This reporter was previously used in studies of defects in these processes in ALS-related models (Zhang et al, Nature, 2015). However, this assay did not reveal major changes in the TDP-43 KO cells. The precise relationship between nuclear envelope morphology and nuclear pore function is likely to be complex and may involve compensatory mechanisms that have allowed the KO cells to adapt and survive. Perhaps long-lived cells such as neurons might be a better model for teasing out the functional consequences of subtle defects in the balance between nuclear import and export. However, the development of new models and assays to investigate this topic is beyond the scope of this study.

5) Are morphological and localization defects of organelles caused by defects in the MT or actin cytoskeleton?

This is an interesting question. Some of the phenotypes that we observe (such as organelle distribution and Golgi fragmentation) fit with predictions for the consequences of microtubule depolymerization. We have therefore performed new experiments to visualize microtubules and F-actin. However, we did not uncover major defects in the TDP-43 KO cells. This data is now presented in Supplemental Figure 2A and B.

6) Can expression of Nup188 rescue morphological and potential functional defects?

Although we identified Nup188 as a novel TDP-43 target gene, our RNA-seq experiments document significant changes in the expression of hundreds of genes in the TDP-43 KO cells. This includes other previously reported TDP-43 target genes with connections to the nuclear envelope such as RANBP1. Although we highlight the impact on Nup188 due to its potential interest to other investigators who are studying nucleocytoplasmic transport defects in ALS, we do not make strong claims about specific cause-and-effect relationships between Nup188 and

any of the other phenotypes. Given the large number of TDP-43-dependent gene expression changes that we have observed, it is not practical to unambiguously establish the functional impacts arising from changes in the expression of individual genes.

7) Rescue experiments with mutant TDP-43 are of limited value. It is unlikely that as the authors state, "as some TDP-43 disease causing mutants failed to support the regulation of specific target transcripts, our results raise the possibility of mutation-specific loss-of-function contributions to disease pathology." This would be of more interest if nuclear localization of TDP-43 were preserved in these patients. However, mutant TDP-43 in fALS still leads to loss of TDP-43 from the nucleus and cytoplasmic aggregation.

The TDP-43 KO cells provide a new model for testing the basic functionality of disease-causing TDP-43 mutants. Beyond the field of neurodegenerative disease, this is relevant for understanding the normal cellular functions of TDP-43. Furthermore, as the loss of nuclear TDP-43 occurs in only a fraction of cells and only late in the disease course, mutant forms of TDP-43 could still have impact on nuclear functions of TDP-43 in fALS patients. By showing that the TDP-43 KO cells can be used to assess the functionality of TDP-43 mutants, we provide a new tool to the community of researchers who are interested in this topic.

Reviewer #2

We thank the reviewer for their overall positive evaluation of our manuscript.

1) What are the splicing changes detected in the TDP-43 transcript that are highlighted in Fig.4B?. The authors might also want to describe a little more this experiment by mentioning how many of these events belong to skipped exons, cryptic splice site activation, intron retention, etc.

TDP-43 was identified as a hit in the analysis of transcript changes due to the fact that the transgene that was used for the rescue experiments did not contain the endogenous 5' and 3' UTR sequences rather than a change in splicing. The figure legend has been updated to explain the origin of this data point. We have updated the description of this data in the legend for Figure 4 to reflect that we investigated changes in individual "transcript abundance" rather than a detailed analysis of the types of "splicing changes". We have not performed further analysis to classify the different types of changes to transcripts/isoforms as such analyses would not fit with the work flow which led to the findings that we have studied in more detail such as the regulation of Nup188 by TDP-43. As such, performing these additional analyses would distract from the focus of our manuscript.

2) Additional information should be provided with regards to the 34 nt cryptic exon inserted in the Nup188 transcript following TDP-43 deletion. Is this exon predicted to introduce the presence of a premature stop codon in the mature mRNA sequence?. If so, then the drop in transcript expression could be due to activation of the NMD pathway. Has this been tested by treating the cells with an NMD inhibitor?.

The insertion of a 34 nucleotide cryptic exon results in a frameshift and the presence of a premature stop codon in exon 4 (out of 44). This is therefore predicted to result in nonsense mediated decay of this transcript and is consistent with the reduced overall levels of Nup188 in the TDP-43 KO cells. This prediction has not been tested with an NMD inhibitor. We have updated the text to more directly communicate these ideas.

3) Authors should note that the difference between the splicing profile of mouse vs. human SORT1 was explained by the presence of differential splicing regulators in its exon 17b (Mohagheghi F et al., 2016).

Thank-you for reminding us of this study which we have now cited. Although these differences between the regulation of mouse and human SORT1 have been worked for several years now, we have received questions on this topic and it is important to communicate this point.

4) It is rather strange that mutations which impair splicing efficiency (ie. K263E) can rescue Nup188 expression that, according to the results of the author, derives from the inclusion of a cryptic exon. It would be very interesting to know whether in the cells that stably express the K263E mutant the cryptic exon is not excluded any more. Has this been tested?. In general, at least for the two examples that deal with splicing regulation (Nup188 and POLDIP3) the authors should also add a quantification of the different splicing isoforms in addition to the Western blots.

We have performed new RT-PCR experiments to more directly assess Nup188 splicing in the cells that were rescued with the K263E mutant (Figure 5 F-H). The results show that the K263E mutant is fully capable of suppressing inclusion of this cryptic exon in Nup188.

The fact that the pattern of functionality that we obtained in these rescue experiments with disease-linked TDP-43 mutants did not fit with expectations based on current literature convinced us that these results should be of interest to other investigators who are working on this topic.

Reviewer #3

We appreciate the thoughtful suggestions and positive evaluation provided by Reviewer 3.

1. This study would significantly benefit from transcriptomics of at least one or two key disease mutant lines compared to the KO studies presented. While a detailed mechanism of why TDP43 mutants fail to completely rescue KO cells is beyond the scope of this paper, it would be helpful to know to what degree mutants can rescue function using a scalable technique. Additional forward-thinking discussion regarding possible mechanisms would also be helpful. For example, we now know that mutant RNA granule proteins can alter the properties of other RNA granule proteins through dysregulated phase separation properties of multi-protein RNPs. How much of the observed lack of rescue is due to pure loss of function of TDP-43 versus dysregulated biology of other members of an RNP due to dysregulated droplet biology?

These are valuable suggestions for new projects that could build on the results that we present in this manuscript. We have added some new discussion about the implications of TDP-43 mutants on phase separations. However, we also note that these questions and suggestions touch on topics that range from transcriptomics to splicing regulation to the biophysics of phase separations that support RNP function. Pursuing all of these new experiments in detail could potentially form the basis for several manuscripts and would involve a timeframe and budget beyond what is feasible for the revisions to the current manuscript.

2. It would be helpful to test what are the predominant TDP43 isoforms in HeLa cells (perhaps these data are even available on online databases). There is at least one additional major TDP-43 isoform that has been described (D'Alton, RNA, 2015; Seyfried et al Mol Cell Proteomics 2010) that lacks the NLS, and recent unpublished data presented at national meetings suggests that this may be a major isoform in vulnerable neurons. Since the rescue studies done here only utilize the full-length isoform, loss of function in neurons that express multiple isoforms may be more complicated.

In the revised manuscript, we have performed anti-TDP-43 immunoblots using antibodies that recognize distinct epitopes in the N and C terminus of the protein. These experiments revealed only a single specific band at 43 kDa (Supplemental Figure S1). We have focused on this isoform and have shown it to be fully functional via rescue experiments. The shorter TDP-43 variant reported by Seyfried et al, was not detectable in these experiments. This isoform furthermore lacks the C-terminal low complexity domain that harbors many of the disease causing mutants that we have studied.

3. The cathepsins are described in their relation to lysosomal localization in the TDP-43 KO line, but this data is not formally shown. For example, Figure 3B shows punctate Cathepsin D, but does not demonstrate that these structures are lysosomes. This interesting result of distinct mechanisms for Cathepsin delivery should be formally shown perhaps with colocalization of Cathepsin D and LAMP1.

This is an important point. We provide new data (Supplemental Figure 3 A-C) showing colocalization of these lysosomal proteins with LAMP1.

4. The authors state in Figure 3C that progranulin delivery to lysosomes is diminished. While the cytoplasmic buildup of progranulin in the TDP-43 KO is clear in the image, it is less clear if localization to lysosomes is actually diminished (or if the cytosolic accumulation merely alters the differential between punctate and cytoplasmic signal). This should be tested perhaps by quantifying total fluorescence signal and/or colocalization to lysosomes.

We now provide LAMP1 colocalization data (Supplemental Figure 3 A-C) to address this concern.

5. The authors show that TDP-43 loss results in both a defect in Nup188 splicing and abnormal staining of Nup62, suggesting a defect in general nuclear pore organization. This result should be verified with staining of an additional nuclear pore marker to distinguish between a general defect or a specific interaction between Nup188 and Nup62 localization. Alternatively, a test of nuclear pore function could be performed to strengthen the modest RANBP1 result reported.

We have performed new experiments wherein we used the NLS-NES tdTomato reporter that was previously used (Zhang et al, Nature, 2015) to measure nuclear import defects in C9-linked ALS patient iPSC-derived neurons. After careful quantification of multiple experiments, we did not detect any difference in the nuclear/cytoplasmic distribution of this reported in control versus TDP-43 KO HeLa cells. The challenge in measuring functional consequences of altered nuclear pore complex/nuclear envelope morphology may reflect robust adaptive mechanisms that maintain homeostasis of this critical cellular organelle.

Minor comments:

1. On page 5 of the introduction, the authors state "These results suggest that loss-of-function arising from cytoplasmic accumulation and aggregation of mutant TDP-43 proteins rather than a direct loss of their fundamental splicing functions causes them to confer disease risk." However, the authors only test the ability of TDP-43 disease mutants to splice a few specific splice targets. Furthermore, they find that mutants can selectively lose the ability to splice specific targets. As such their data would also be consistent with TDP-43 mutants contributing to disease via a loss of splicing activity of one or more unidentified splice targets. They also do not examine cytosolic accumulation or aggregation of the mutant proteins expressed in the TDP-43 KO background to back this claim. We suggest revising the text to take a more agnostic view point.

We are grateful for this suggestion and have revised this paragraph accordingly.

2. For Figure 4 that authors describe an insertion of a cryptic 34 bp exon in Nup188 in the TDP-43 knockout lines. While the authors describe in the methods that the transcriptomics were done in triplicate, it would be interesting to know if the cryptic exon was identical between samples or if there was some variability in the absence of TDP-43 activity.

We observed comparable insertion of the cryptic 34 bp exon in Nup188 between all 3 pairs of samples that were analyzed by RNA-seq. In figure 4, we furthermore present evidence that this cryptic exon is immediately followed by a series of UG repeats that conform to the previously defined TDP-43 consensus binding site.

August 15, 2019

RE: Life Science Alliance Manuscript #LSA-2019-00358R

Dr. Shawn Michael Ferguson
Yale School of Medicine
Department of Cell Biology
295 Congress Ave BCMM254E
New Haven, CT 06510

Dear Dr. Ferguson,

Thank you for submitting your revised manuscript entitled "Pleiotropic requirements for human TDP-43 in the regulation of cell and organelle homeostasis". As you will see, the reviewers appreciate the introduced changes and we would thus be happy to accept your manuscript for publication in Life Science Alliance, pending final revisions to meet our formatting guidelines:

- Please upload all figures (including S figures) as individual files and without figure legends, the legends for S figures and S tables should go into the main manuscript text file, Table S3 and S4 can also go into the main manuscript file
- Please provide your manuscript file in word docx format
- Please add callouts in the manuscript text to Fig 1C and Fig S6A
- Please indicate the origin of the magnification boxes in Fig S3
- Please deposit the RNA-seq data and provide the accession code

A. FINAL FILES:

B. MANUSCRIPT ORGANIZATION AND FORMATTING:

Sincerely,

Reviewer #2 (Comments to the Authors (Required)):

Authors have made some significant changes to the manuscript and as a result it has been noticeably improved compared to the original submission. Not all the requested experiments were added but reasonable explanations were provided. In conclusion, the descriptive data will be of interest to the community

Reviewer #3 (Comments to the Authors (Required)):

The authors have completed numerous additional experiments at the request of the reviewers, including those requested by me. These data further strengthen an already compelling story, and I now fully support its publication.

September 2, 2019

RE: Life Science Alliance Manuscript #LSA-2019-00358RR

Dr. Shawn Michael Ferguson
Yale School of Medicine
Department of Cell Biology
295 Congress Ave BCMM254E
New Haven, CT 06510

Dear Dr Ferguson,

Thank you for submitting your Research Article entitled "Pleiotropic requirements for human TDP-43 in the regulation of cell and organelle homeostasis". It is a pleasure to let you know that your manuscript is now accepted for publication in Life Science Alliance. Congratulations on this interesting work.

DISTRIBUTION OF MATERIALS:

Again, congratulations on a very nice paper. I hope you found the review process to be constructive and are pleased with how the manuscript was handled editorially. We look forward to future exciting submissions from your lab.